SlZIP11 mediates zinc accumulation and sugar storage in tomato fruits

Sun Jiaqi 1
Wang Manning 1
Zhang Xinsheng 2
Liu Xin 1 3
Jiang Jing 1 3 jiangj_syau@syau.edu.cn
1 College of Horticulture, Shenyang Agricultural University , Shenyang, Liaoning , China
2 College of Horticulture, Jilin Agricultural University , Changchun, Jilin , China
3 Key Laboratory of Protected Horticulture of Education Ministry , Shenyang, Liaoning , China
Irfan Mohammad
Electronic publication date: 2024 May 29
Publication date: 2024
Volume: 12
Electronic Location ID: e17473
Received 2024 Feb 8; Accepted 2024 May 6
Copyright: © 2024 Sun et al.
Copyright year: 2024
Copyright holder: Sun et al.
License: This is an open access article distributed under the terms of the Creative Commons Attribution License, which permits unrestricted use, distribution, reproduction and adaptation in any medium and for any purpose provided that it is properly attributed. For attribution, the original author(s), title, publication source (PeerJ) and either DOI or URL of the article must be cited.
License URL: https://creativecommons.org/licenses/by/4.0/

Keywords: Tomato, SlZIP11, Fruit, Zinc content, SlSWEET7a, Sugar transport

Funding: Science and Technology Program of Liaoning province 2022020768-JH1/102–02 to Jing Jiang This study was supported by grants from the Science and Technology Program of Liaoning province (2022020768-JH1/102–02 to Jing Jiang). The funders had no role in study design, data collection and analysis, decision to publish, or preparation of the manuscript.

==============================
Background

Zinc (Zn) is a vital micronutrient essential for plant growth and development. Transporter proteins of the ZRT/IRT-like protein (ZIP) family play crucial roles in maintaining Zn homeostasis. Although the acquisition, translocation, and intracellular transport of Zn are well understood in plant roots and leaves, the genes that regulate these pathways in fruits remain largely unexplored. In this study, we aimed to investigate the function of SlZIP11 in regulating tomato fruit development.

Methods

We used Solanum lycopersicum L. ‘Micro-Tom’ SlZIP11 (Solanum lycopersicum) is highly expressed in tomato fruit, particularly in mature green (MG) stages. For obtaining results, we employed reverse transcription-quantitative polymerase chain reaction (RT-qPCR), yeast two-hybrid assay, bimolecular fluorescent complementation, subcellular localization assay, virus-induced gene silencing (VIGS), SlZIP11 overexpression, determination of Zn content, sugar extraction and content determination, and statistical analysis.

Results

RT-qPCR analysis showed elevated SlZIP11 expression in MG tomato fruits. SlZIP11 expression was inhibited and induced by Zn deficiency and toxicity treatments, respectively. Silencing SlZIP11 via the VIGS technology resulted in a significant increase in the Zn content of tomato fruits. In contrast, overexpression of SlZIP11 led to reduced Zn content in MG fruits. Moreover, both silencing and overexpression of SlZIP11 caused alterations in the fructose and glucose contents of tomato fruits. Additionally, SlSWEEET7a interacted with SlZIP11. The heterodimerization between SlSWEET7a and SlZIP11 affected subcellular targeting, thereby increasing the amount of intracellularly localized oligomeric complexes. Overall, this study elucidates the role of SlZIP11 in mediating Zn accumulation and sugar transport during tomato fruit ripening. These findings underscore the significance of SlZIP11 in regulating Zn levels and sugar content, providing insights into its potential implications for plant physiology and agricultural practices.

Introduction

Zinc (Zn) is an important micronutrient that acts as a co-factor for important enzymes in plants (Broadley et al., 2007). The disruption of metabolic processes due to Zn deficiency reduces crop growth and productivity. Therefore, maintaining Zn homeostasis through various uptake and translocation strategies is crucial for plants. Zn is absorbed by the roots primarily as divalent cations (Zn2+) and transported to the xylem through the symplast and apoplast routes (Gupta, Ram & Kumar, 2016).

To overcome low Zn availability, plants have evolved strategies to cope with Zn deficiency. Zn transporters, particularly members of the Zn-regulated transporters (ZRT)/iron regulated transporter (IRT)-like protein (ZIP) family, play essential roles in maintaining Zn2+ balance in plants. They also play key roles in the uptake, root-to-shoot translocation, sequestration, and distribution of Zn (Noémie & Marc, 2021). In Arabidopsis, 15 ZIP family proteins have been characterized as Zn transporters, and most of them are induced by Zn deficiency (Van de Mortel et al., 2006). For instance, AtZIP1 and AtZIP2, which are located on the vacuolar membrane and plasma membrane, respectively, mediate Zn transport from the roots to the shoots. They also promote transportation of Zn from the central pillar to the xylem parenchyma, where the xylem is loaded and transported to the shoots (Milner et al., 2013). The Zn transporter NtZIP11 is highly expressed in Nicotiana tabacum leaves and contributes to the accumulation of Zn in the leaves in cases of excessive presence of Zn (Kozak et al., 2018). VvZIP3 plays a role in the absorption and distribution of Zn during early reproductive development in grapes (Vitis vinifera) (Gainza-Cortés et al., 2012). Notably, Zn deficiency can induce the expression of several ZIPs (Ramesh et al., 2003; Ishimaru et al., 2005). For example, NtZIP4B is upregulated under conditions of Zn deficiency and downregulated under excessive levels of Zn (Barabasz et al., 2019).

“Sugars Will Eventually be Exported Transporter” (SWEET) proteins have recently been identified as sugar transporters (Chen et al., 2010, 2012). Plant sugar uptake transporters (SUTs) are regulated by oligomerization or protein interaction. Hetero-oligomerization between SUTs in yeast could reduce their sucrose transport activity (Reinders et al., 2002), whereas homo-oligomerization of StSUT1 has predominately been observed in phloem sieve elements. Both StSUT1 and StSUT4 have been shown to form heterodimers mainly in the endoplasmic reticulum (Liesche et al., 2010). Considering the reported regulatory mechanisms of other sugar transporters, the regulation of SWEET proteins may occur at the post-translational level. In a previous study, split-ubiquitin yeast two-hybrid and split green fluorescent protein (GFP) assays indicated that Arabidopsis SWEET proteins form homo- or hetero-oligomeric complexes (Xuan et al., 2013). Furthermore, SWEET proteins can interact with ion transporters to regulate ion transport and distribution in plants (Yuan et al., 2010). Rice Xa13/Os8N3/OsSWEET11, which localizes to the plasma membrane, can interact with plasma membrane-localized copper transporter (COPT) 1 and COPT5 to promote the removal of copper from xylem vessels in rice (Yuan et al., 2010).

Tomato is a high-value crop with fruit as its product organ. Enhancing the Zn content in plant-based foods could help in addressing global human malnutrition. Until now, a total of nine Zn transporters gene family members, including low-affinity (SlZIP5L2, SlZIP5L1, SlZIP4, and SlZIP2) and high-affinity members (SlZIPL), have been identified in the tomato genome (Pavithra et al., 2016). In Zn deficiency-sensitive cultivar Ratan, SlZIPL and SlZIP3 are not upregulated, while reduction in biomass and photosynthetic activity as well as cell death are detectable after 1 week of Zn deficiency (Akther et al., 2020; Aslam et al., 2020). The levels of Zn transporter-like protein (Solyc07g065380) and Zn transporter protein (Solyc06g005620) were significantly reduced following Zn-deprivation in either the roots or shoots of tomato (Pavithra et al., 2016). However, the effects of Zn transporter activity on the physiological and molecular aspects of tomato fruits remain largely unknown. SlZIP11 (Solyc08g066500) is highly expressed during the ripening stage of tomato fruit; however, its role in the regulation of fruit development is elusive. Accordingly, in this study, we aimed to investigate the function of SlZIP11 gene in regulating tomato fruit development. To this end, SlZIP11-overexpressing (OE-ZIP11) and silenced (TRV-ZIP11) lines were generated. TRV-ZIP11 lines promoted Zn accumulation and sugar storage, while in OE-ZIP11 lines, both Zn accumulation and sugar storage were reduced. Moreover, we found that SlZIP11 interacted with SlSWEEET7a. The alteration in Zn accumulation and sugar storage in SlZIP11 transgenic fruits might be due to the interaction between SlZIP11 and SlSWEET7a. Our findings provide important insights into the role of ZIPs and SWEET proteins in the regulation of sugar content and Zn levels in tomato fruits. The results of this study will provide insights into important genes for regulating the content of Zn and sugar in tomato fruit as well as target genes for genetic improvement of tomato nutritional quality and biological breeding.

Materials and Methods

Plant materials

Tomato plants (Solanum lycopersicum L. ‘Micro-Tom’, MT) and transgenic variants derived from this cultivar were grown in growth cabinets at 25 °C with a 16/8 h photoperiod and a relative humidity of 60–70% (Cheng et al., 2018). Mature green (MG) and red ripening fruits (RR) were collected from at least three plants 35 and 55 days after anthesis (DAA), respectively. Samples were immediately frozen in liquid nitrogen and stored at −80 °C until use.

Reverse transcription-quantitative polymerase chain reaction (RT-qPCR)

TRIzol reagent (Tiangen, Beijing, China) was used to extract total RNA from plant tissues according to the manufacturer’s instructions. A reverse transcription kit (HiScript II Q RT SuperMix forqPCR (+gDNA wiper); Vazyme, Nanjing, China) was used to synthesize double-stranded cDNA according to the manufacturer’s instructions. A SYBR real-time fluorescence quantitative kit (ChamQ Universal SYBR qPCR Master Mix; Vazyme) was used for RT-qPCR. RT-qPCR assays were performed as described previously (Feng et al., 2015). The tomato housekeeping gene ACTIN was used as an internal control.

Yeast two-hybrid assay

Tomato cDNA was used as the template to amplify SlZIP11 and SlSWEET7a target fragments with the primers listed in Table S1. The coding (CDS) sequence of SlZIP11 without the ATG codon and stop codon was cloned into the pBT3-STE bait vector using the SfiI site, and the CDS sequence of SlSWEET7a with the ATG codon and stop codon was introduced into the pPR3-N prey vector using the SfiI site. Recombinant vectors were introduced into yeast strain NMY51. The assays were performed as described previously (Zhang et al., 2019). All experiments were performed at least thrice, and representative results are presented.

Bimolecular fluorescent complementation

Gateway technology was used to construct bimolecular fluorescence complementation vectors. Micro-Tom cDNA was used as the template to amplify the SlZIP11 and SlSWEET7a target fragments with the primers listed in Table S1. The CDS sequences (without stop codons) of SlZIP11 and SlSWEET7a were cloned into the pXNGW and pXCGW vectors, respectively (Xuan et al., 2013). The fusion proteins were cloned into Agrobacterium tumefaciens strain GV3101, and the bacterium was then used to infect Nicotiana benthamiana leaves. The fluorescence signals were detected 2 days later with a confocal laser scanning microscope (Leica SP8; Germany) as previously described (Zhang et al., 2021a). All experiments were performed at least thrice, and representative results are presented.

Subcellular localization assay

The cDNA of MT was used as a template to amplify the SlZIP11 target fragment with the primers listed in Table S1. The CDS sequence of SlZIP11 without the stop codon was introduced into the pCAMBIA1302-GFP vector at the KpnI site. The resulting recombinant vector was introduced into A. tumefaciens strain GV3101, which was then used to infect N. benthamiana leaves. The fluorescence signals were detected 2 days later with a confocal laser scanning microscope as described previously (Zhang et al., 2021a). All experiments were performed at least thrice, and representative results are presented.

Virus-induced gene silencing (VIGS)

The 300-bp fragment of SlZIP11 for gene silencing was selected via the VIGS tool on the Sol Genomics Network (https://solgenomics.net/) and cloned into the pTRV2 vector at the KpnI site using the primers listed in Table S1. TRV1, TRV2, and pTRV2-ZIP11 were cloned into A. tumefaciens strain GV3101. The transformants were grown overnight at 28 °C in LB medium (containing 50 mg/L kanamycin, 50 mg/L rifampicin, 10 mM MES, and 20 μM acetosyringone). The cells were collected and suspended in an infiltration buffer (10 mM MgCl2, 10 mM MES, and 200 μM acetosyringone) to obtain an optical density at 600 nm of 1.0. 3 h later, a mixture of equivalent cultures was infiltrated into the cotyledons of 1-week-old tomato plants with a 1 mL syringe at room temperature. Uniformly sized plants were used for infiltration, and the experiment was repeated thrice. A total of 2 weeks after infiltration, RT-qPCR was employed to detect the efficiency of silencing.

Transgenic tomato plants overexpressing SlZIP11

The CDS sequence of SlZIP11 without the stop codon was introduced into the pCAMBIA3301-Luc vector at the XbaI and XmaI sites. The resulting pCAMBIA3301-pro35S-SlZIP11 overexpression vector was cloned in A. tumefaciens strain GV3101 and further transferred into MT using the leaf disc method (Guo et al., 2012). Positive plants were selected based on phosphinothricin (found in glufosinate-ammonium, a herbicide) resistance and PCR analysis of the T1 generation. The T2 generation was used for functional studies.

Determination of Zn content

Pooled samples were dried in a drying oven and ground into a powder. The powder (0.1 g) was placed in a digestion bottle, supplemented with 4 mL of a nitric acid-perchloric acid mixed acid solution, and digested at 200 °C until the solution was converted to a colorless, transparent liquid. After cooling, the sample (25 mL) was transferred to a volumetric flask containing ultrapure water. Zn content in the digestion liquid was determined using an atomic absorption spectrophotometer.

Extraction and determination of sugar content

For determining the sugar content, two identical tomato fruits were collected from one plant and mixed into one sample, followed by three biological. Samples (0.5 g) of MG (35 DAA) and RR fruits (55 DAA) of WT (wild type), OE-ZIP11, TRV, and TRV-ZIP11 transgenic plants were collected. Subsequently, fruit samples were added to glass test tubes containing 80% ethanol (25 mL) and placed in a water bath (°C) to extract sugar as described previously (Zhang et al., 2020). Sucrose, glucose, and fructose contents were analyzed via HPLC on a 600E HPLC system (Waters, Milford, MA, USA) equipped with an amino column (Dikma Technologies, Inc., Foothill Ranch, CA, USA); the 2410 differential detector was used. Data were processed using Waters Millennium software v. 32 (Waters, Milford, MA, USA) (Zhang et al., 2021a).

Data analysis and statistics

For each experiment, three biological replicates and three technical replicates were used. Significant differences among the treatments and control were determined using a one-way ANOVA and SPSS Statistics 17.0 software (IBM Corp., Armonk, NY, USA). Standard error of the mean was calculated and is indicated in the graphs by an error bar. Asterisks (∗ and ∗∗) indicate a significant difference between the controls and transgenic plants at p < 0.05 and p < 0.01.

Results

Expression pattern of SlZIP11 under Zn treatment

We identified a membrane-tethered Zn transporter, NP_001234349.1 (Solyc07g065380.2). Phylogenetic tree analysis of the protein sequence of the Zn transporter with 15 ZIP family members in Arabidopsis showed that Solyc07g065380.2 had the highest homology with AtZIP11 in Arabidopsis; therefore, we named the protein as SlZIP11 (Fig. 1).

Figure 1 Phylogenetic analysis of the Zn transporter.

Bootstrap values indicate the confidence of each branch, and the scale indicates branch length.

Furthermore, the expression pattern of SlZIP11 was assessed in different tissues of MT during different developmental stages (Fig. 2A). The results showed that SlZIP11 was highly expressed during tomato fruit development, especially in MG. Moreover, SlZIP11 was highly expressed in the leaves and branches. The tomato database website was used to obtain data about SlZIP11 expression pattern in cultivated tomatoes (Fig. S1). The result was similar to that in MT. SlZIP11 was significantly expressed during the fruit expansion stage.

Figure 2 Expression analysis of SlZIP11 in Micro-Tom (MT) fruits.

(A) Relative expression patterns of SlZIP11 in different tissues of MT tomato during different development stages. IMG: immature green fruits (10 days post-anthesis (DPA)); MG: mature green fruits (35 DPA); BR: breaker fruits (38 DPA); RR: red ripening fruits (44 DPA). Expression data of SlZIP11 in the roots was normalized to 1. The ACTIN gene was used as the internal reference. (B) Expression levels of SlZIP11 in MT leaves and MG after Zn treatment, including without ZnSO4 (-Zn: Zn deficient), with 2 mM ZnSO4 (control: Zn sufficient) and with 100 mM ZnSO4 (Zn toxicity). The ACTIN gene was used as the internal reference. Data are presented as the mean ± SE of values from three independent biological replicates. Significant differences compared to the control were determined using Student’s t-test at *p < 0.05, **p < 0.01.

To determine whether the expression of SlZIP11 is affected by Zn treatment, we investigated the expression of SlZIP11 in MT leaves and MG under Zn treatment, including without ZnSO4 (-Zn: Zn deficient), with 2 mM ZnSO4 (control: Zn sufficient), and with 100 mM Zn (Zn toxicity). Results showed that in the leaves, SlZIP11 was significantly downregulated under Zn-deficient treatment compared to that in CK. However, under Zn toxicity, SlZIP11 was significantly upregulated, approximately twice more than that in CK (Fig. 2B). Regarding MG, the change in SlZIP11 expression was less significant than that in the leaves compared to that in CK. Furthermore, SlZIP11 in MG was also significantly down- and upregulated under Zn deficiency and toxicity, respectively, compared with that in CK. Overall, these results showed that SlZIP11 might function in tomato fruit development and is affected by Zn availability.

SlZIP11 is located on the plasma membrane

Pro35S::SlZIP11::GFP fusion was used to confirm the subcellular localization of SlZIP11. The recombinant vector was introduced into Agrobacterium GV3101 for subsequent infiltration into N. benthamiana leaves. AtPIP2A (a plasma membrane-located protein) was used as the plasma membrane marker. Upon the co-expression of SlZIP11-GFP with AtPIP2A-mCherry in tobacco leaves, green and red fluorescence signals were observed around the cells and were located on the plasma membrane (Fig. 3). Therefore, we inferred that SlZIP11 is located on the plasma membrane.

Figure 3 Subcellular localization of SlZIP11 in N. benthamiana.

AtPIP2A-mCherry was used as the plasma membrane marker. The empty vector was used as the positive control. The experiment was performed thrice independently.

Silencing and overexpression of SlZIP11 alter Zn content in tomato fruits

To further assess the function of SlZIP11 in tomato fruits, we constructed silencing and overexpressing lines of SlZIP11 (OE) (Figs. 4A, 4C). The expression of SlZIP11 in TRV-ZIP11 was significantly reduced (approximately 55–80%) compared to that in the TRV control lines (Fig. 4A). Furthermore, in TRV-ZIP11 lines, the Zn content in MG fruits was significantly increased (approximately 10%) compared to that in the control lines (Fig. 4B).

Figure 4 Zn content in the fruits of transgenic plants.

(A) Expression levels of SlZIP11 in TRV-ZIP11 plants. (B) Zn content in the mature green (MG) fruits of TRV-ZIP11 and TRV plants. (C) Expression levels of SlZIP11 in OE-ZIP11 plants. (D) Zn content in the MG fruits of OE-ZIP11 and CK plants. Significant differences compared to the control were determined using Student’s t-test at *p < 0.05, **p < 0.01.

Regarding OE-ZIP11 lines, the expression of SlZIP11 was 5- to 10-fold higher than that in CK (Fig. 4C). In the OE-ZIP11 MG fruits, there was an approximate 50% decrease in Zn content compared to that in CK (Fig. 4D). These results showed that SlZIP11 was involved in Zn accumulation in tomato fruits during the MG stage.

Changes in the expression of SlZIP11 are associated with the difference in accumulation of sugar in tomato fruits

Given the high expression of SlZIP11 in tomato fruits, we investigated whether SlZIP11 participates in sugar accumulation in tomato fruits by measuring the soluble sugar content in TRV-ZIP11 MG and RR fruits. In the MG fruits, the fructose and glucose contents in TRV-ZIP11 were significantly increased compared to those in the TRV lines (Fig. 5A). In RR fruits, the fructose and glucose contents in TRV-ZIP11 were also increased; however, the sucrose content in TRV-ZIP11 was reduced compared with that in the TRV control fruits (Fig. 5B).

Figure 5 Fruit sugar content in SlZIP11 transgenic plants.

(A and B) Sugar content in the fruits of TRV-ZIP11 and TRV plants at the mature green (MG) (A) and red ripening (RR) (B) stages. (C and D) Sugar contents of OE-ZIP11 and CK fruits at the MG (C) and RR (D) stages. Significant differences compared to the control were determined using Student’s t-test at *p < 0.05, **p < 0.01.

We also measured the sugar content in OE-ZIP11 lines. Results showed that the fructose and glucose contents in OE-ZIP11 MG fruits were increased, whereas the sucrose content in OE-ZIP11 was reduced in MG fruits compared with that in CK fruits (Fig. 5C). In RR fruits, the contents of fructose and glucose in OE-ZIP11 were significantly increased compared to those in CK, with a simultaneous decrease in sucrose content (Fig. 5D). These findings indicate that alterations in the transcript levels of SlZIP11 affect sugar accumulation in tomato fruits during the MG and RR stages.

SlZIP11 interacts with a membrane-tethered SlSWEET7a transporter

We previously identified SlSWEET7a as a functional sugar transporter that regulates sugar transport in tomato fruits (Cheng et al., 2018; Zhang et al., 2021a). To elucidate its underlying mechanism of action, we screened for potential SlSWEET7a interactors using a split-ubiquitin membrane yeast two-hybrid (MYTH) system. We used the split-ubiquitin-based MYTH and bimolecular fluorescence complementation (BiFC) to demonstrate that SlSWEET7a interacts with SlZIP11 in vivo and in planta, respectively (Fig. 6). Yeast transformants were plated onto SD medium lacking leucine and tryptophan (SD/-Trp/-Leu+X-gal) and SD medium lacking histidine, leucine, tryptophan, and adenine, and containing 3-amino-1,2,3-triazole (SD/-Trp/-Leu/-His/-Ade+3-AT). On the SD/-Trp/-Leu/-His/-Ade+3-AT medium, only PBT3-STE-SlSWEET7a with pPR3N-SlZIP11 yeast transformants grew; the control did not (Fig. 6A). In BiFC experiments, interacting green fluorescence was clearly observed on the plasma membrane and intracellularly for mixed injections of SlSWEET7a-nYFP and SlZIP11-cCFP or SlZIP11-nYFP and SlSWEET7a-cCFP in the green fluorescence channel. In the merged field, an overlap between the green fluorescence and the cell membrane of tobacco cells was detected, demonstrating that the two proteins interacted in the cell membrane and that heterodimer formation seemed to increase SlSWEET7a internalization (Fig. 6B).

Figure 6 SlSWEET7a interacts with SlZIP11.

(A) Split-ubiquitin yeast two-hybrid assay revealing interactions between SlSWEET7a and SlZIP11. Interactions were tested using the His and Ade reporter genes and 3-amino-1, 2, 4-triazole (3-AT) (10 mM) (right panel) and verified using an X-gal (100 μg mL–1) staining assay (left panel). pR3N-SlZIP11 with pBT3-STE and pR3N with pBT3-STE-SlSWEET7a were used as the negative controls. (B) Interaction of SlSWEET7a with SlZIP11 in a bimolecular fluorescence complementation assay. SlSWEET7a-nYFP+SlZIP111-cCFP is shown in the upper panel, and SlZIP11-nYFP+SlSWEET7a-cCFP is shown in the lower panel. SlZIP11nYFP with cCFP and SlSWEET7a--nYFP with cCFP were used as the negative controls. Agrobacterium-mediated transient expression of the indicated constructs in N. benthamiana leaves. Reconstitution of GFP-derived fluorescence and bright field and merged images are shown in the left, middle, and right panels, respectively. The green signal indicates GFP fluorescence. Scale bars correspond to 25 μm.

Discussion

Interaction of SlZIP11 and SlSWEET7a regulates Zn accumulation and sugar transport in tomato fruits

Recent studies have illuminated various post-translational mechanisms that regulate sucrose transporters, including processes such as phosphorylation, oligomerization, protein–protein interactions, and subcellular redistribution (Krügel & Kühn, 2013). Such post-translational control has been observed for SUC2-interaction partners in Arabidopsis. SUC2 activity is regulated via its protein turnover rate and phosphorylation state (Xu et al., 2020). Additionally, ubc34 mutants show increased phloem loading as well as increased biomass and yield (Zhang et al., 2021b). Mutants of another SUC2-interaction partner, wall-associated kinase-like 8 (WAKL8), show reduced phloem loading and growth. An in-vivo assay based on a fluorescent sucrose analog confirmed that SUC2 phosphorylation by WAKL8 could increase transport activity (Ma et al., 2022). In tomatoes, screening of an expression library for SlSUT2-interacting proteins revealed interactions of SlSUT2 with elements of brassinosteroid biosynthesis and signaling, with both involved in the regulation of mycorrhizal symbiosis (Bitterlich et al., 2014). Studies on SWEET family proteins have already been reported in Arabidopsis and rice, demonstrating that these proteins can occur in homologous or hetero-multimerized forms to perform sugar transport functions (Xuan et al., 2013).

In previous studies, we found that SlSWEET7a is specifically expressed in tomato fruits, particularly during the MG stage (Cheng et al., 2018; Zhang et al., 2021a). In the study, MYTH and BiFC analyses confirmed that the two proteins interacted with each other. The results of SlZIP11 gene silencing and overexpression experiments showed that the fructose and glucose contents of fruits in the silenced and overexpressing lines increased and that the red fruit sugar content in the overexpressing line increased significantly. These findings indicate that SlZIP11 may interact with SlSWEET7a to regulate the transport and metabolism of sugar in fruits.

A few MtN3/saliva/SWEET-type genes have been shown to be associated with ion transport or involved in ion homeostasis (e.g., aluminum, NH4+, and boron) in plants (Zhao et al., 2009; Lopes & Araus, 2008). For instance, rice Xa13 (OsSWEET11) interacts with two homologs of the COPT family of copper transporters to participate in the redistribution of copper in rice and is related to the removal of copper in the xylem (Antony et al., 2010). Co-expression of three copper transport proteins—COPT1, COPT5, and OsSWEET11—was shown to restore copper uptake defects in the copper transport-deficient yeast mutant MPY17 (Antony et al., 2010). Similarly, our results indicate an interaction between SlSWEET7a and the Zn transporter SlZIP11. The ZIP family transporters regulate Zn availability, and their expression patterns vary in different tissue systems. SlZIP4 and SlZIP5 are required for Zn transport in the shoot and root systems (Rahman et al., 2024). In the current study, SlZIP11 expression was found to be higher in the fruits than in the other parts. Furthermore, our results showed that SlZIP11 may interact with SlSWEET7a to regulate the transport and metabolism of sugar in fruits. This approach may improve fruit quality by regulating the activity of Zn transporters via modulation of SWEET protein expression and subsequent control of Zn distribution in plants.

Interaction between SlSWEET7a and SlZIP11 affects targeting to the plasma membrane

The results of the analysis of subcellular localization of SlZIP11 showed that the protein was localized in the plasma membrane. Notably, heterodimerization between SlSWEET7a and SlZIP11 was found to affect subcellular targeting. Indeed, heterodimerization resulted in increased amount of intracellularly localized oligomeric complexes compared with that of the monomeric forms of SlSWEET7a and SlZIP11, which were mainly localized at the plasma membrane. Similar to intracellularly localized oligomers, SlSUT2-interacting proteins also showed interactions with other transporters internalized via endocytosis, such as AUX1 or various proton ATPases, or with components of the vesicle trafficking machinery (Bitterlich et al., 2014). However, further investigations are required to understand the exact mechanism of interaction between SlSWEET7a and SlZIP11 during the fruit development and ripening processes. In addition, whether this inhibition of sucrose uptake is related to the increased internalization of SlSWEET7a by heterodimerization should be clarified.

Conclusions

Plasma membrane-located protein SlZIP11 mediates Zn accumulation and sugar transport by interacting with SlSWEET7a during tomato fruit ripening.

Supplemental Information

Supplemental Information 1 List of primer sequences and expression level of SlZIP11 in ‘Heinz 1706’.

IMG_10DPA, immature green fruits (10 day post-anthesis); MG_35DPA, mature green fruits (35DPA); Breaker_38DPA, breaker fruits (38DPA); Orange_41DPA, orange fruits (41DPA); Red_44DPA, red ripen fruits (44DPA). The data was obtained from the tomato database (https://solgenomics.net/)

Supplemental Information 2 MIQE checklist.

Supplemental Information 3 Expression analysis of SlZIP11 in Micro-Tom (MT) fruit.

Relative expression pattern of SlZIP11 in different tissues of MT tomato during different development stages. IMG: immature green fruits (10 day post-anthesis); MG: mature green fruits (35DPA); BR: breaker fruits (38DPA); RR: red ripening fruits (44DPA). The expression data of SlZIP11 in root was normalized to 1. The ACTIN gene was used as the internal reference. (B)The expression level of SlZIP11 in MT leaves and MG after Zn treatment including without ZnSO4 (-Zn: Zn deficient), 2 μM ZnSO4 (control: Zn sufficient) and 100 μM ZnSO4 (Zn toxicity). The ACTIN gene was used as the internal reference. Mean values ± SE of three independent biological replicates are given. Significance compared to control was determined by Student’s t test at *P<0.05, **P<0.01.

Supplemental Information 4 Zinc content in fruits of transgenic plants.

The expression level of SlZIP11 in ZIP11 transgenic and CK plants. Zinc content in MG fruits of ZIP11 transgenic and CK plants. Signifificance compared to control was determined by Student’s t test at *P<0.05, **P<0.01.

Supplemental Information 5 Fruit sugar content in SlZIP11 transgenic plants.

Sugar content in fruits of TRV-ZIP11 and TRV plants at MG and RR stage. Sugar content in OE-ZIP11 and CK fruits at MG and RR stage. Signifificance compared to control was determined by Student’s t test at *P<0.05, **P<0.01.

Additional Information and Declarations

Competing Interests

Author Contributions

Data Availability

The authors declare that they have no competing interests.

Jiaqi Sun conceived and designed the experiments, performed the experiments, analyzed the data, prepared figures and/or tables, authored or reviewed drafts of the article, and approved the final draft.

Manning Wang conceived and designed the experiments, performed the experiments, analyzed the data, prepared figures and/or tables, authored or reviewed drafts of the article, and approved the final draft.

Xinsheng Zhang conceived and designed the experiments, performed the experiments, authored or reviewed drafts of the article, and approved the final draft.

Xin Liu conceived and designed the experiments, authored or reviewed drafts of the article, and approved the final draft.

Jing Jiang conceived and designed the experiments, authored or reviewed drafts of the article, and approved the final draft.

The following information was supplied regarding data availability:

The raw measurements are available in the Supplemental File.

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
