# Peer review of "SlZIP11 mediates zinc accumulation and sugar storage in tomato fruits"

_PeerJ, doi:10.7717/peerj.17473_

## Round 0.1 · original submission · Major Revisions

· Academic Editor

Major Revisions

Your manuscript was reviewed by three experts in the field. The reviewers found the work interesting but raised several issues which need to be addressed properly. The reviewers provide detailed comments in their reviews and point out the areas where the manuscript needs to be improved. I also read the manuscript carefully and largely agree with the reviewers’ comments.

Reviewer 3 has requested that you cite specific references. You may add them if you believe they are especially relevant. However, I do not expect you to include these citations, and if you do not include them, this will not influence my decision.

**Language Note:** The review process has identified that the English language must be improved. PeerJ can provide language editing services - please contact us at [email protected] for pricing (be sure to provide your manuscript number and title). Alternatively, you should make your own arrangements to improve the language quality and provide details in your response letter. – PeerJ Staff

Reviewer 1 ·

Basic reporting

The manuscript presents a clear and well-structured investigation into the role of SlZIP11 in zinc accumulation and sugar storage in tomato fruits. The English used is professional throughout, facilitating easy understanding. However, there are a few grammatical errors and typos that need correction. The literature references provided offer sufficient background context for the study. Figures and tables are well incorporated, aiding in the visualization of the results. Raw data is shared in supplementary files, enhancing the transparency and reproducibility of the study.

Experimental design

The research question regarding the role of SlZIP11 in zinc accumulation and sugar storage in tomato fruits is well-defined and meaningful, filling an identified knowledge gap in the field. The investigation is conducted to a high technical and ethical standard, with rigorous methods described in sufficient detail to allow for replication. However, there are some inconsistencies and ambiguities in the experimental procedures that need clarification, such as the exact methods used for sugar extraction and determination of concentration.

Validity of the findings

The impact and novelty of the findings are not explicitly assessed in the manuscript. Encouragement for meaningful replication is present, and the rationale for such replication is clear. All underlying data have been provided, contributing to the robustness and transparency of the study. The conclusions drawn are well-stated and linked to the original research question, supported by the results obtained. However, the discussion could benefit from further elaboration on the potential implications and significance of the findings in the broader context of plant biology and agricultural practices. Additionally, it is advisable to italicize gene/TF names for consistency and clarity.

Reviewer 2 ·

Basic reporting

1. English language is okay, however repeated sentences have been used. Attachment between two words should be avoided throughout the manuscript.
2. Literature references seem to be sufficient.
3. Article structure look reasonably good.
4. Article is self-contained with relevant results.

Experimental design

1.Present article shows original primary research within Aims and Scope of the journal.

2.The present manuscript clearly defines the research question, which are relevant and meaningful.

3. I want authors to look into their statistical analysis whether they are significant or not?

4. Methodes have been described with sufficient details and information to replicate however, I am curious whether two fruits per plant were enough to collect relevant information on Sugar, starch, and Zn ion concentrations?
Concentrations of Zn were given in micromolar in article and milli molars in raw data. Correction required..

Validity of the findings

1. Present research is not novel.

2. Authors should reconsider their statistical data analysis.

3. Conclusions are well stated, linked to original research question & limited to supporting results. However, few conclusions are entirely literature based.

Additional comments

1. OE-is the right abbreviation for over-expression?
2. Gaps and word to word distance should be thoroughly checked in the present article.
3. There is no experimental evidence of plant disease resistance in present article.

Reviewer 3 ·

Basic reporting

This article is good read. Minor English changes required. Sufficient background provided.

Experimental design

The research question is well defined, relevant, and significant.

Validity of the findings

Finding are important in the field of zinc transporter and sugar storage in tomato fruits.

Additional comments

The article entitled, “The SlZIP11 mediates zinc accumulation and sugar storage in tomato fruits" was evaluated by this reviewer. Researchers examined a gene SlZIP11 from tomato that exhibited high expression levels in tomato fruit. Zn transporters are essential for maintaining the balance of Zn2+ in plants. They utilized reverse transcription-quantitative PCR and revealed increased expression of SlZIP11 in mature green tomato fruits. As well as Zn deficiency or toxicity treatment reduced or increased SlZIP11 expression. Moreover, Silencing SlZIP11 using VIGS technology significantly boosted zinc concentration in tomato fruit. Additionally, overexpression or silencing of SlZIP11 altered the fructose and glucose concentration in tomato fruits.

In conclusions they proposed SlZIP11 mediates Zn accumulation and sugar transport during tomato fruits ripening.

Altogether this is an important and timely article, this reviewer has certain suggestions that would help produce a more comprehensive overview of the topic:

Comments:
1, At least one additional Figure (illustration) may be provided as to highlight the summary or prospect of this study.
2, The English of manuscript can be polished (minor) and there are few typological errors.
3, Authors should provide limitations to their study.
4, I would suggest adding few citations to put comprehensive view of this topic (PMID: 35842058, PMID: 38294556, PMID: 34866296 etc).

---

## Round 0.2 · Minor Revisions

· Academic Editor

Minor Revisions

I appreciate authors effort in revising the manuscript as suggested by the reviewers. The current version looks better, however, I noticed several typographical and English language errors throughout the manuscript including the title ('mediate' should be 'mediates'). Authors need to check them thoroughly.

Reviewer 1 ·

Basic reporting

The revised manuscript is in acceptable mode.

Experimental design

The revised manuscript is in acceptable mode.

Validity of the findings

The revised manuscript is in acceptable mode.

---

## Round 0.3 · accepted · Accept

· Academic Editor

Accept

Authors have addressed all the concerns raised during the review process.